# Evaluating Cardiac Lateralization by MRI to Simplify Estimation of Cardiopulmonary Impairment in Pectus Excavatum

**DOI:** 10.3390/diagnostics13050844

**Published:** 2023-02-23

**Authors:** Tariq Abu-Tair, Salmai Turial, Ines Willershausen, Muhannad Alkassar, Gundula Staatz, Christoph Kampmann

**Affiliations:** 1Department of Congenital Heart Disease, Centre for Diseases in Childhood and Adolescence, University Medicine Mainz, 55131 Mainz, Germany; 2Department of Pediatric Cardiology, Friedrich-Alexander-University Erlangen-Nürnberg, 91054 Erlangen, Germany; 3Department of General, Visceral, Vascular and Transplantation Surgery, Division of Pediatric Surgery, Pediatric Trauma Surgery and Pediatric Urology, University Medical Centre Magdeburg, 39120 Magdeburg, Germany; 4Department of Orthodontics and Orofacial Orthopedics, Friedrich-Alexander University Erlangen-Nürnberg, 91054 Erlangen, Germany; 5Department of Diagnostic and Interventional Radiology, Section of Pediatric Radiology, University Medicine Mainz, 55131 Mainz, Germany

**Keywords:** pectus excavatum, cardiopulmonary exercise testing, chest wall deformity, Haller index, correction index, cardiac lateralization

## Abstract

Background: The severity of pectus excavatum is classified by the Haller Index (HI) and/or Correction Index (CI). These indices measure only the depth of the defect and, therefore, impede a precise estimation of the actual cardiopulmonary impairment. We aimed to evaluate the MRI-derived cardiac lateralization to improve the estimation of cardiopulmonary impairment in Pectus excavatum in connection with the Haller and Correction Indices. Methods: This retrospective cohort study included a total of 113 patients (mean age = 19.03 ± 7.8) with pectus excavatum, whose diagnosis was verified on cross-sectional MRI images using the HI and CI. For the development of an improved HI and CI index, the patients underwent cardiopulmonary exercise testing to assess the influence of the right ventricle’s position on cardiopulmonary impairment. The indexed lateral position of the pulmonary valve was utilized as a surrogate parameter for right ventricle localization. Results: In patients with PE, the heart’s lateralization significantly correlated with the severity of pectus excavatum (*p* ≤ 0.001). When modifying HI and CI for the individual’s pulmonary valve position, those indices are present with greater sensitivity and specificity regarding the maximum oxygen-pulse as a pathophysiological correlate of reduced cardiac function (χ^2^ 10.986 and 15.862, respectively). Conclusion: The indexed lateral deviation of the pulmonary valve seems to be a valuable cofactor for HI and CI, allowing for an improved description of cardiopulmonary impairment in PE patients.

## 1. Introduction

Pectus excavatum (PE), defined as the posterior positioning of the sternum and costal cartilages, is the most frequent congenital anterior chest wall deformity, with an incidence of 1 in 300–400 births and an m:f ratio of 2:1 [1,2]. While mild forms of PE are treated conservatively, moderate to severe types usually require surgery [3,4]. Patients with severe PE suffer from aesthetic, psychosocial, and cardiopulmonary impairments, e.g., dyspnea, chest discomfort, palpitations and exercise limitations; therefore, they usually undergo minimally invasive surgery in early adolescence [5]. The pathophysiology of the cardiopulmonary impairment in PE patients can be explained by the compression of the right ventricle, which leads to a decreased diastolic filling of the right ventricle, resulting in both reduced stroke and right ventricular ejection fraction [6,7,8,9,10]. The most important parameter for a description of cardiopulmonary impairment in patients with PE is the maximum stroke volume of the right ventricle. The oxygen-pulse, which can be measured non-invasively by a Cardiopulmonary Exercise Test (CPET), is considered as a surrogate parameter for the right ventricle’s stroke volume [11]. PE patients present with greater inter-individual variability with respect to severity, exercise status, impairment, and compensatory mechanisms [9,12]. The heart’s lateralization has been previously described as a pathophysiological consequence of chest wall intrusion in patients with PE without correlating it to the effective cardiac dysfunction. The respective eccentricity of the heart is usually classified by a centroid calculation of the maximum cross-sectional area, or by the indexed lateral cardiac border to the right and left side of the midline [10,13,14]. However, no correlation between cardiac lateralization and resulting cardiopulmonary impairment has been described. We assume that the varying degree of cardiac dysfunction in patients with moderate and severe types of PE might be partially attributed to the heart’s lateral displacement as a result of the chest wall intrusion [10,13,15]. While a rather central heart position might lead to a pronounced compression of the right ventricle, with reduced ventricular ejection fraction and stroke-volume, a lateral displacement may, consequently, reduce its compression and minimize patients’ cardiopulmonary impairment [16]. In addition to the Haller and Correction index, multiple indices exist, such as the Depression index, the Sternal Torsion index, and the Titanic index, that can be used to describe the morphology of the funnel chest [17,18,19,20,21,22]. Several publications relate PE to cardiac compression using the aforementioned indices, which have led to the development of other indices, such as the Cardiac Compression Index or the Cardiac Asymmetry Index, which require the collection of at least four additional variables [8,16,23,24]. The Haller Index and Correction Index are the two most commonly used indices for the classification of chest wall deformities; however, they do not take the aforementioned anatomical variations into consideration [20,21,22]. An assessment of the influence of the severity of chest-wall intrusion on cardiopulmonary impairments for varying anatomical heart positions has been published as an MRI study, but this lacked cut-off values regarding when a finding is to be classified as pathological and when it is accompanied by a reduction in performance [23]. We aimed to establish modifications for HI and CI, since they can be easily obtained from cross-sectional imaging. These improved indices should provide both information on the morphological severity of PE and the degree of the right ventricle’s lateral displacement as a cofactor for an estimation of cardiopulmonary impairment. Since the lateral position of the pulmonary valve is rather stable during the heart cycle due to its predominantly axial systolic–diastolic movement, it was used as a calculation value for right ventricular displacement in combination with HI and CI, which are used as surrogate parameters for cardiac compression [25,26].

The aim of this case-control study was to adjust HI and CI to cardiac lateralization, using this as a cofactor to improve the determination of cardiopulmonary impairment in cross-sectional imaging.

## 2. Materials and Methods

### 2.1. Patients

We retrospectively analyzed young adults and adolescents presenting with pectus excavatum who received surgery for PE at the department of pediatric surgery, Centre for Diseases in Childhood and Adolescence of the University Medicine Mainz, Germany. MRI was advised and CPET was performed by the Department of Congenital Heart Disease, University Medicine Mainz, Germany. The institutional review board waived the need for approval, since the data were obtained due to medically necessary routine examinations. However, all legal guardians and adult patients provided written informed consent for the scientific use of their radiological and clinical data. From September 2012 to January 2019, 113 patients with PE (m = 97, f = 16, mean age = 19.03 ± 7.8) underwent CPET and thoracic MRI in hold end-inspiration, both of which were performed within 1 week. Nine additional patients who were diagnosed with additional cardiac, pulmonary, and neurologic diseases, leading to impaired CPET results, were excluded prior to enrollment in this study. A total of 36 otherwise healthy subjects (m = 22, f = 14, mean age = 19.06 ± 2.35) who received an MRI on account of long-term follow-up after lymphoma therapy in childhood were used as controls regarding the position of the right ventricle, HI_ins_, and CI_ins_ (Figure 1).

All patients underwent MRI scans on a 1.5-Tesla (T) scanner (MAGNETOM Avanto, Siemens AG, Munich, Germany) or a 3-T scanner (MAGNETOM Skyra, Siemens AG, Munich, Germany) with a gradient strength of 45 mT/m and a 32-channel body coil for signal detection. At the University of Mainz, pre-operative PE-diagnostic enclosed MR-imaging. A standard T2-weighted half-Fourier acquisition single-shot turbo spin echo sequence with an acquisition time of 25–30 s in hold end-inspiration was used to calculate the HI_insp_ and CI_insp_, as well as the modHI and modCI. Both the lateral position of the pulmonary valve, reaching from the midline of the chest to the middle of the pulmonary valve (a) and the diameter of the inner chest wall at the level of the pulmonary valve (b), were measured to calculate the indexed position of the pulmonary valve (Figure 2).

The lateral position of the pulmonary valve (a) measured from the midline (sterno-spinal line) to the middle of the pulmonary valve and the inner thoracic diameter at the same level (b). Haller Index and Correction Index were calculated in standard fashion (Sectra Imaging Software Version 24.1).

The indexed pulmonary valve position was calculated by dividing the distance of the center of the pulmonary valve from the midline according to half the internal thoracic diameter at the level of the pulmonary valve.
$$\mathrm{Indexed}\;\mathrm{pulmonary}\;\mathrm{valve}\;\mathrm{Position}=\frac{\mathrm{Distance}\;\mathrm{of}\;\mathrm{pulmonary}\;\mathrm{valve}\;\mathrm{from}\;\mathrm{midline}}{\mathrm{thoracic}\;\mathrm{diameter}\;\mathrm{at}\;\mathrm{level}\;\mathrm{of}\;\mathrm{pulmonary}\;\mathrm{valve}/2}$$

The subject depicted above shows the pulmonary valve located from the midline of 4.88 cm, with an inner thoracic diameter at the pulmonary valve level of 23.66 cm. This results in an indexed pulmonary valve position of 0.41. The indexed pulmonary valve position serves as a reciprocal cofactor for the calculation of the modHI and modCI.
$$\mathrm{modified}\;\mathrm{Haller}\;\mathrm{Index}=\frac{\mathrm{Haller}\;\mathrm{Index}}{\mathrm{indexed}\;\mathrm{pulmonary}\;\mathrm{valve}\;\mathrm{position}}$$
$$\mathrm{modified}\;\mathrm{Correction}\;\mathrm{Index}=\frac{\mathrm{Correction}\;\mathrm{Index}}{\mathrm{indexed}\;\mathrm{pulmonary}\;\mathrm{valve}\;\mathrm{position}}$$

The HI of 4.34 results in a modHI of 10.59, and the CI of 35.67 results in a modCI of 87 are shown.

### 2.2. CPET Procedure

CPET was performed on an H/P/cosmos treadmill using CareFusion MS-CPX 5.70 software (CareFusion, Hoechberg, Germany) combined with GE-electrocardiogram (ECG) Cardiosoft 6.7 (GE Healthcare, Chicago, IL, USA). The exercise protocol was an altered, improved treadmill protocol recommended by the German Society of Pediatric Cardiology. According to this protocol, the test started with at 2 km/h on a flat treadmill, increasing stepwise in increments of 0.5 km/h and 3% inclination every 90 s to a maximum incline of 21%. Beyond this inclination, only speed continued to increase, in increments of 0.5 km/h. The testing was stopped when the calculated target heart rate (i.e., 220–age in years) was reached or when exhaustion occurred, as indicated by the patient. The percentage of predicted oxygen uptake (VO_2max_% [%]) and percentage of predicted oxygen pulse (O_2_-pulse_max_% [%]) were evaluated and compared against calculated reference values based on age, sex, height, and weight. Cardiopulmonary impairment in CPET was defined as a reduction in VO_2max_% of less than 85% of the predicted value and a reduction in oxygen-pulse of less than 80% of the predicted value.

### 2.3. Statistics

Statistical analyses were conducted using the SPSS 24 software (IBM, Armonk, NY, USA). The Mann–Whitney U test was used to compare age, ratio of pulmonary valve position, HI_insp_, CI_insp_, modHI, and modCI of PE patients with the control group values. HI_insp_ and CI_insp_ were set in relation to indexed pulmonary valve position, and R^2^ was calculated. After dividing patients in quartiles according to HI_insp_ and CI_insp_, a Kruskal–Wallis Test was performed to compare the quartiles according to pulmonary valve position index.

Receiver operating characteristic (ROC) analysis was performed to evaluate the sensitivity and specificity of the four different indices in relation to the defined cardiopulmonary impairment. Unpaired Student *t*-tests were performed to compare CPET results at different grades of severity based on modHI and modCI.

Additionally, a Kruskal–Wallis Test was performed to compare CPET results with modHI and modCI after dividing pectus patients into quartiles. Contingency tables were calculated at different grades of modHI and modCI to evaluate the positive predictive value, negative predictive value, sensitivity, and specificity of the defined cardiopulmonary impairment. Significance was determined by *X*^2^-Test. A *p*-value of less than 0.05 was considered significant in all statistical analyses.

## 3. Results

Our patient collective comprised a total of 113 patients with PE (m = 97, f = 16; mean age = 19.03 ± 7.8) and 36 age-matched controls (m = 22, f = 14; mean age = 19.06 ± 2.35). Further biometric data of the entire collective are given in Table 1.

The position of the pulmonary valve was significantly more lateral in pectus patients than in the control group (*p* < 0.001). Further, a significant difference between the PE and the control group regarding HI_insp_, modHI, CI_insp_, and modCI could be established, with less favorable results being found in the PE collective. After dividing the entire collective into quartiles according to HI_insp_ and CI_insp_, a Kruskal–Wallis Test was performed to compare these quartiles regarding the pulmonary valve position index. The indexed position of the pulmonary valve differed significantly between the quartiles of HI_insp_ and/or CI_insp_ values (*p* ≤ 0.001). The data and distribution are depicted in Figure 3.

To discriminate between pectus patients and healthy controls, modCI should be utilized rather than modHI. In modCI, the overlap between the PE and control group is only 12%, as opposed to 94% in the modHI group. Cardiopulmonary impairment is defined as a VO_2max_% less than 85% of the predicted value and/or O_2_-pulse_max_% less than 80% of the predicted value. Both values are related to HI_insp_, modHI, CI_insp_, and modCI.

All indices depicted below displayed a cardiopulmonary impairment with regard to a reduced VO_2max_% (ROC analyses: HI_insp_, *p* = 0.004, CI_insp_, *p* = 0.017, modHI, *p* = 0.012, modCI, *p* = 0.01). Only modHI and modCI showed a highly significant difference regarding their reduced O_2_-pulse_max_% (ROC analyses: modHI, *p* ≤ 0.001, modCI, *p* ≤ 0.001). The O_2_-pulse_max_% reduction did not reach a level of significance in HI_insp_ and CI_insp_ (ROC analyses: HI_insp_, *p* = 0.116, CI_insp_, *p* = 0.061). ROC curves are depicted in Figure 4.

The sensitivity, specificity, positive predictive value (PPV), and negative predictive value (NPV) of modHI and modCI used in the detection of VO_2max_% were reduced to less than 85%, while those for O_2_-pulse_max_% were reduced to less than 80%. These can be seen in the results of the contingency tables, given in Table 2. Additionally, the Chi-square test was utilized to evaluate the highest level of significance and highest χ^2^ according to the reduced cardiopulmonary function, as described by reduced VO_2max_% or reduced O_2_-pulse_max_%.

A total of 22 probands had a VO_2max_% of less than 85% of the predicted value and a total of 24 patients had an O_2_-pulse_max_% of less than 80% of the predicted value. The sensitivity and specificity in the detection of impaired cardiopulmonary function according to a reduced VO_2max_% for modHI were 92% and 40% at an index of 12. For modCI, the sensitivity and specificity were 77% and 57%, respectively. Both had a high negative predictive value of 91% and 90%. The sensitivity and specificity in the detection of impaired cardiopulmonary function according to a reduced O_2_-pulse_max_% for modHI were 92% and 40% at an index of 12; for modCI, they were 71% and 74%, respectively, at an index of 120. Both had a high negative predictive value of 94% and 89%. An unpaired *t-*test revealed a significantly reduced O_2_-pulse_max_%, reaching maximum statistical significance at a modHI of 12, with a reduction of 101.75 ± 17.94% to 89.89 ± 18.90% (*p* = 0.003). The VO_2max_% was reduced, reaching its maximum statistical significance at a modHI of 13 with a reduction from 101.3 ± 13.83% to 92.56 ± 14.85% (*p* = 0.003). O_2_-pulse_max_% was reduced, reaching its maximum statistical significance at a modCI of 120 with a reduction from 98.95 ± 18.49% to 84.78 ± 17.56% (*p* ≤ 0.001), and VO_2max_% was significantly reduced, reaching its maximum statistical significance for a reduced VO_2max_% at 100.

Regarding modHI, a statistical significance could be observed between the quartiles in VO_2max_% (*p* = 0.024) and in O_2_-pulse_max_% (*p* = 0.002). The same analysis regarding modCI revealed statistically significant differences between the quartiles in VO_2max_% (*p* = 0.029) and O_2_-pulse_max_% (*p* = 0.016). Data for VO_2max_% and O_2_-pulse_max_%, according to quartiles of modHI and modCI, are depicted in Figure 5.

## 4. Discussion

Although the Haller and Correction indices are the two most commonly used indices for the classification of chest wall deformities, they do not adequately describe the actual cardiopulmonary function in patients with PE [11,21,22,27]. In patients with pectus excavatum incongruences between severity levels, classified by defect depth according to HI and CI, respective cardiopulmonary impairment is frequently observed [11,28]. In addition to general cardiac impairment, there is only one publication describing HI and CI impairment as pathologic. In this, a reduced VO_2max_% and O_2_-pulse_max_% occurred with rather low sensitivity and specificity [11]. Some patients whose defects were classified as severe presented with a normal cardiopulmonary function, while cases with moderate pectus excavatum showed severe cardiopulmonary impairment [8,23]. Explanatory approaches alluded to in the literature are compensatory mechanisms, such as an increased heart rate, different exercise status, and deviating grades of right ventricular compression. The findings of this study underline that the position of the right ventricle, as measured on MRI-scans, correlates with the severity of the pectus excavatum; however, there are marked inter-individual variations, a phenomenon which has been described in the previous literature [14]. An anatomical rationale for this observation is that the compression of the right ventricle is reduced in a more lateral position of the heart. Hence, a reduced right ventricular function, which is found in patients with a median heart position, leads to greater cardiopulmonary impairment than a heart with a more lateral position [29,30]. Therefore, we indexed the position of the pulmonary valve and used it as a surrogate parameter for the right ventricle’s position. We then modified HI and CI accordingly, which allowed for a more accurate estimation of the actual cardiopulmonary impairment. We evaluated the heart’s lateralization as a cofactor in the morphology, which is depicted in the established indices and modified HI and modified CI. These new modified indices, which were adjusted for the heart’s lateralization, suggest an improved possibility for the assessment of the actual cardiopulmonary impairment using cross-sectional imaging. This leads to a precise allocation of the pathophysiological substrate; hence, the reduced O_2_-pulse_max_% shows severe PE. We recommend using the oxygen pulse as an indicator for cardiopulmonary impairment, since it is an accepted surrogate parameter for stroke volume while being independent from heartrate [31]. The heartrate is the most important compensatory mechanism in pectus excavatum, since a reduction in maximum oxygen uptake is outbalanced by an increased heartrate, and this might lead to inaccuracies [11]. In previous studies evaluating autonomal dysfunction, a reduced heartrate level and heartrate variability were noted by Holter-ECG [32,33]. The outbalancing of a reduced oxygen-pulse due to an increased heart rate, which allowed for normal VO_2max_% to be achieved in PE, could be further evaluated in Holter-ECG-Studies. The modified HI and CI are easy to obtain using cross-sectional imaging and can be calculated by acquiring only two variables (thorax diameter and position of the pulmonary valve from the midline). We utilized MRI (T2w-HASTE-sequence) to measure thoracic deformities without radiation and focused on the inner thoracic areas with a short acquisition time of 25–30 s and adequate spatial resolution [34].

Previous studies described significant differences in the HI-index when measured during in- and expiration [35,36]. As a consequence, we recommend that the MRI scans should be performed during end-inspiration, since only this setting adequately reflects the cardiopulmonary function and its respective impairment [11]. When looking at the exercise testing, the CPET parameters revealed a significant decrease in O_2_-pulse_max_% and VO_2max_% at an MRI-derived inspiratory modified HI, with a value ranging from 11.5 to 12. O_2_-pulse_max_% and VO_2max_% were continuously reduced in all modified CI, reaching their maximum statistical significance at a modified CI of 120 and 100, respectively. These findings suggest a possible link between the degree of PE, lateralization, and reduced cardiopulmonary function. When considering that both modified indices show a respective overlap between pectus patients and healthy controls, we recommend the use of HI or CI for the initial diagnosis and morphologic description of pectus excavatum. The presented modifications can also be used to estimate the grade of cardiopulmonary impairment with a high degree of sensitivity and specificity (Table 2). There are many publications regarding the cardiopulmonary limitations due to pectus excavatum. A variety of measurement parameters have been collected to determine these from cross-sectional imaging. However, most studies offer little indication of when an index leads to pathological cardiopulmonary exercise capacity, e.g., a reduced VO_2max_% or O_2_-pulse_max_%. These are defined in the literature using threshold values [31,37]. Due to the large number of studies, indices, and measurement methods, a combination of studies using data envelopment analysis will be necessary in the future in order to optimize the diagnostic and therapy planning based on multiple variables. Artificial intelligence models and hybrid parametric and non-parametric optimization models can also be used for this purpose [38,39,40,41,42]. Finally, all parameters collected in multiple studies must be evaluated to determine whether there is a normalization of cardiac dysfunction and alterations after PE correction.

## 5. Limitations

Our study has several limitations, which are mainly based on the retrospective design and small sample size. At our department, PE is routinely verified by MRI; however, we are aware that this does not represent the standard in other institutions, particularly in adult patients, who mostly receive CT scans. MRI scans are often believed to be time-consuming and expensive, and are not readily available at every location. The MRI sequence was applied in our study; hence, the T2-weighted, half-Fourier acquisition, single-shot, turbo spin echo sequence has an acquisition-time of 25–30 s and does not need a contrast medium [15,34]. Our methodology can only be also transferred to CT scans with a contrast medium [43] depicting cardiac structures, but offers a much higher resolution within an extremely short acquisition time. Our modified indices are based on the Haller and Correction Indexes, the most-used indices in the diagnosis of PE, which have been supplemented by the cofactor of the lateral position of the pulmonary valve. This cofactor is only a surrogate parameter for the position of the right ventricle and only a surrogate for cardiac compression in combination with HI and CI. Cardiac compression is not directly evaluated in an index such as the cardiac compression index. Therefore, our indices could be considered a valuable addition to the preexisting indices; however, they are not recommended for use on their own. Recent cardiac and morphological indices provide an improved morphological description of PE, which is suitable for surgical planning, but the chest wall intrusion is sufficiently described in the Haller and Correction indices. These well-established indices are sufficient for the diagnosis of PE, and our modification could be considered for the additional evaluation of cardiac impairment, especially in countries where PE repair is acknowledged as a cosmetic surgery by insurance companies.

## 6. Conclusions

The findings of the present study display modifications to HI and CI for the lateral position of the pulmonary valve. These allow for an accurate correlation between the severity of pectus excavatum and the respective cardiopulmonary impairment. These indices can be readily obtained from any cross-sectional imaging and easily integrated in the preoperative workflow of PE.

## Figures and Tables

**Figure 1 diagnostics-13-00844-f001:**
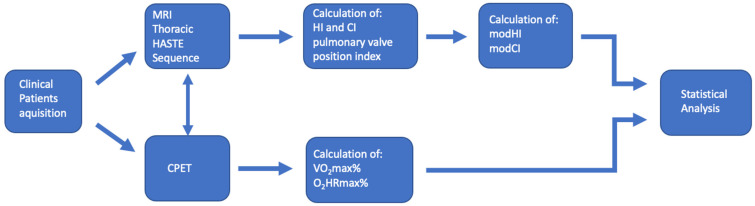
Displays a flowchart of the study design for data acquisition and the calculation of modified HI and CI.

**Figure 2 diagnostics-13-00844-f002:**
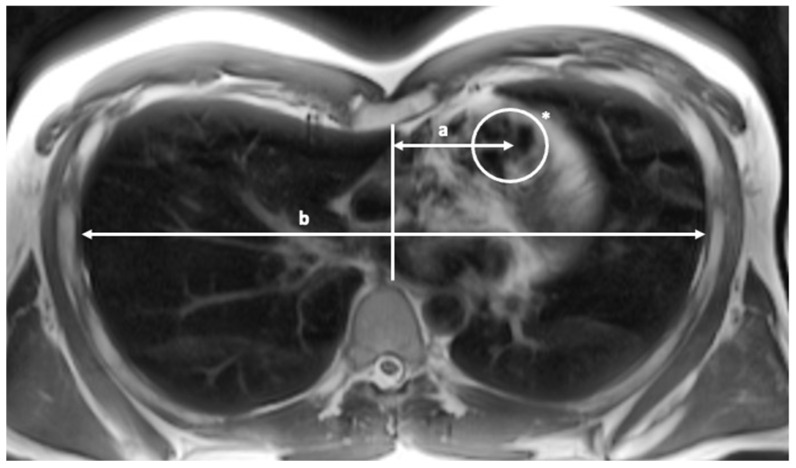
Indexed pulmonary valve position. * = marked the pulmonary valve ring.

**Figure 3 diagnostics-13-00844-f003:**
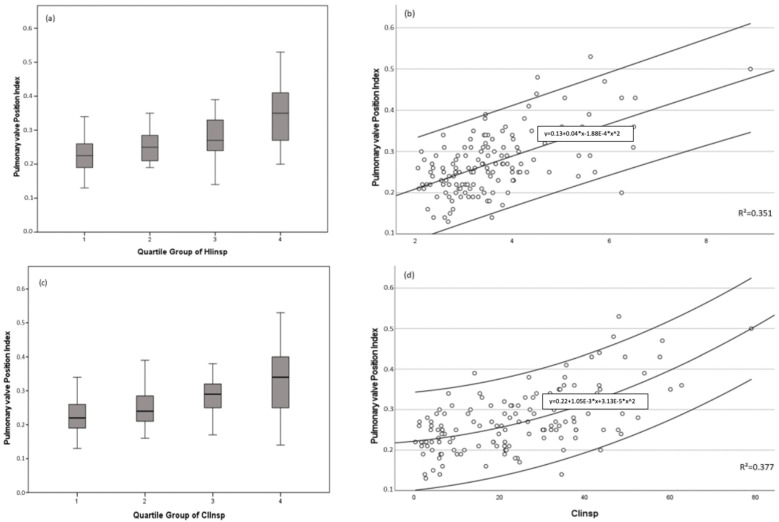
Indexed pulmonary valve position in relation to quartiles of HIinsp and CIinsp. (**a**,**b**) depict the distribution of indexed pulmonary valve position according to HI_insp_; (**c**,**d**) show the distribution of indexed pulmonary valve position according to CI_insp._

**Figure 4 diagnostics-13-00844-f004:**
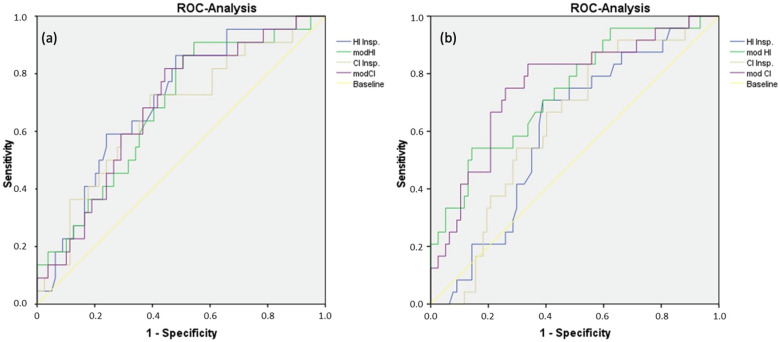
ROC analyses of reduced VO_2max_% and reduced O_2_-pulsemax%. (**a**) shows ROC curves of sensitivity and 1-specificity of HI_insp_ (AUC 0.703), modHI (AUC 0.677), CI_insp_ (AUC 0.667), and modCI (AUC 0.681), related to a VO_2max_% less than 85% as the state variable. (**b**) shows ROC curves of sensitivity and 1-specificity of HI_insp_ (AUC 0.067), modHI (AUC 0.738), CI_insp_ (AUC 0.627), and modCI (AUC 0.757), related to an O_2_-pulse_max_% less than 80% as the state variable.

**Figure 5 diagnostics-13-00844-f005:**
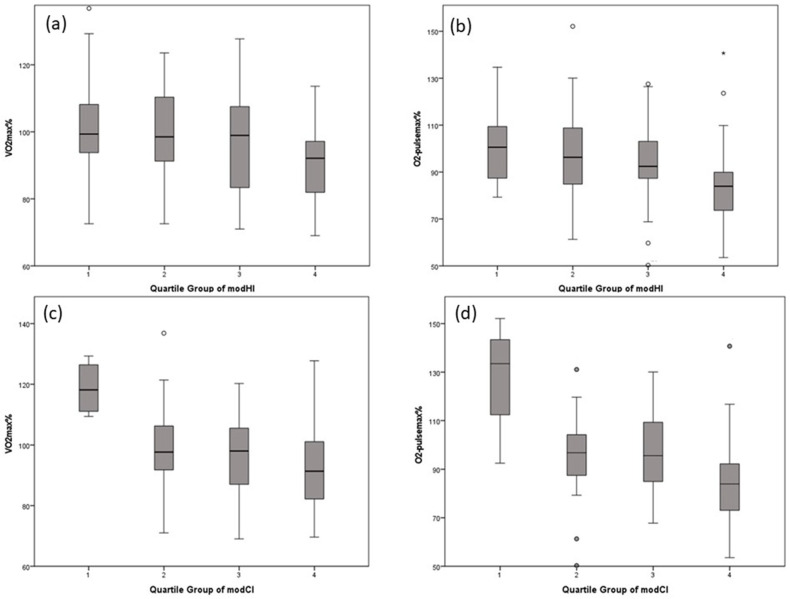
VO_2max_% and O_2_-pulse_max_% according to quartiles of modHI and modCI. (**a**,**b**) depict boxplots of VO_2max_% and O_2_-pulse_max_% according to quartiles of modHI; (**c**,**d**) depict boxplots of VO_2max_% and O_2_-pulse_max_% according to quartiles of modCI.

**Table 1 diagnostics-13-00844-t001:** Biometric and morphometric data according to pectus excavatum and control group.

Biometric/Morphometric Data		Pectus Excavatum	Control	*p*
		N = 113	N = 36	
Age (years)	x ± SD	19.03 ± 7.8	19.06 ± 2.35	0.893
	Median	[16.03]	[18.5]	
	Range	(9.80–44.5)	(16.2–26.4)	
Indexed position of pulmonary valve	x ± SD	0.29 ± 0.08	0.224 ± 0.05	<0.001
	Median	[0.28]	[0.22]	
	Range	(0.14–0.53)	(0.13–0.34)	
HI_ins_	x ± SD	3.97 ± 1.11	2.57 ± 0.29	<0.001
	Median	[3.63]	[2.62]	
	Range	(2.26–8.9)	(2.06–3.08)	
modHI	x ± SD	14.15 ± 3.91	12.15 ± 3.38	0.007
	Median	[13.82]	[11.35]	
	Range	(8.3–31.25)	(7.06–20.49)	
CI_ins_ (%)	x ± SD	30.62 ± 13.39	4.87 ± 2.47	<0.001
	Median	[30.23]	[4.91]	
	Range	(6.67–78.98)	(0.34–9.85)	
modCI	x ± SD	107.07 ± 42.06	22.64 ± 11.94	<0.001
	Median	[101.49]	[21.39]	
	Range	(25.11–244.27)	(1.54–47.66)	

**Table 2 diagnostics-13-00844-t002:** Results of contingency tables to detect impaired cardiopulmonary function.

	VO_2max_%		O_2_-pulse_max_%	
Cut-Off	modHI 13	modCI 100	modHI 16	modCI 120
PPV	32%	33%	46%	46%
NPV	91%	90%	85%	89%
Sensitivity	82%	77%	54%	71%
Specificity	52%	57%	81%	74%
χ^2^	7.919	8.068	10.986	15.862
*p*	0.005	0.005	0.001	<0.001

The contingency tables were calculated at the level of maximum statistical significance in *t*-tests, focusing on the VO_2max_% and O_2_-pulse_max_% when using modHI and modCI.

## Data Availability

T.A.-T. and C.K. had full access to all the data in the study and take responsibility for the integrity of the data and the accuracy of the data analysis. The data presented in this study are available on request from the corresponding author. The data are not publicly available due to privacy (medical data).

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
