# Peer review of "Evaluating Cardiac Lateralization by MRI to Simplify Estimation of Cardiopulmonary Impairment in Pectus Excavatum"

_diagnostics, 2023, doi:10.3390/diagnostics13050844_

Round 1

Reviewer 1 Report

 This paper studies the Evaluating cardiac lateralization by MRI to simplify estimation of cardiopulmonary impairment in Pectus excavatum. The indexed lateral deviation of the pulmonary valve seems to be a valuable cofactor to HI and CI for an improved description of cardiopulmonary impairment in patients with PE.  Therefore, I suggest the authors resubmit it after a major revision. My suggestions are as follows:

1. Discuss the study's limitations and future research suggestions.

2. I strongly suggest that the paper be proofread and reread meticulously again, particularly regarding the spelling and grammatical mistakes.

 3. Flowchart is beneficial; it’s also important to outline the methodology behind this new approach. Please consider a flowchart of your suggested approach at the beginning of your paper.

4. Add a literature review section after the introduction; not as a subsection in the materials and methods part.

 5. Please outline the structure of your paper at the end of the introduction with more details.

 6. Figure 2 is a picture of some formula that is unacceptable. You should write the formula separately with the numbers based on the MDPI structure.

7.  I suggest that you update Table 2 so that the illustration used in the methodology section should be more readable.

 8. Please clarify the definitions for limitations. What is the main strategy behind limitation?

9. It is necessary to include additional information for Figure 5.

10. What is your mean by 329-331? "The modified HI and CI are easy to obtain on cross-sectional imaging depicting cardiac structures, and can be calculated by acquiring only 2 additional parameters (thorax diameter and position of the pulmonary valve from the midline) only"

11. To improve your related works, 30 references are not enough and consider the following related high-quality papers in your literature review:

- Cardiac MRI in the evaluation of acute cardiac toxicity following cancer therapy, with a focus on radiation (Doctoral dissertation, UNSW Sydney).

- Cardiovascular autonomic function and baroreflex sensitivity in drug-resistant temporal lobe epilepsy. Epilepsy & Behavior. 2023 Jan 1; 138:109013.

-  Heart rate and heart rate variability in patients with chronic inflammatory joint disease: the role of pain duration and the insular cortex. BMC Musculoskeletal Disorders. 2022 Dec;23(1):1-2.

-  Cardiac Phase and Flow Compensation Effects on REnal Flow and Microstructure AnisotroPy MRI in Healthy Human Kidney. Journal of Magnetic Resonance Imaging. 2022 Nov 18.

 -  . Comprehensive presurgical functional MRI language evaluation in adult patients with epilepsy. Epilepsy & Behavior. 2008 Jan 1;12(1):74-83.

-  Developing a novel integrated generalised data envelopment analysis (DEA) to evaluate hospitals providing stroke care services. Bioengineering. 2021 Dec 10;8(12):207.

- Cardiovascular computed tomography in pediatric congenital heart disease: A state of the art review. Journal of Cardiovascular Computed Tomography. 2022 May 6.

 -  A novel hybrid parametric and non-parametric optimisation model for average technical efficiency assessment in public hospitals during and post-COVID-19 pandemic. Bioengineering. 2021 Dec 27;9(1):7.

-   Neural correlates of heart-focused interoception: a functional magnetic resonance imaging meta-analysis. Philosophical Transactions of the Royal Society B: Biological Sciences. 2016 Nov 19;371(1708):20160018.

-  An integrated artificial intelligence model for efficiency assessment in pharmaceutical companies during the COVID-19 pandemic. Sustainable Operations and Computers. 2022 Jan 1; 3:156-67.

In conclusion, this version is unacceptable and needs to apply all the suggested comments point by point. Using the suggested high-quality related references is necessary.

Author Response

Dear Reviewer,

thank you very much for your highly constructive comments, we tried to consider all your remarks and found them very helpful to further enhance the quality of this paper.

Thank you very much for your thorough revision.

Please find enclosed our comments.

  1. Discuss the study's limitations and future research suggestions.

Thank you very much for this very constructive comment. We discussed the limitations and future research suggestions in the Discussion- and Limitations-Section according to your recommendations.

  1. I strongly suggest that the paper be proofread and reread meticulously again, particularly regarding the spelling and grammatical mistakes.

The paper received a proof-Read and English language Editing by MDPI.

Certificate: No. 59769

  1. Flowchart is beneficial; it’s also important to outline the methodology behind this new approach. Please consider a flowchart of your suggested approach at the beginning of your paper.

We added a Flowchart according to your recommendation.

  1. Add a literature review section after the introduction; not as a subsection in the materials and methods part.

Literature review has been extended in introduction. References have been removed from materials and methods section and partially integrated in Discussion- and Limitations-Section

  1. Please outline the structure of your paper at the end of the introduction with more details.

This has been done at the end of Introduction-Section

  1. Figure 2 is a picture of some formula that is unacceptable. You should write the formula separately with the numbers based on the MDPI structure.

Formula have been integrated in text and were edited during Layout-Editing by MDPI. Additionally, the patient, depicted in the Figure 2 has been calculated as an example.

  1. I suggest that you update Table 2 so that the illustration used in the methodology section should be more readable.

Has been updated according to your recommendation.

  1. Please clarify the definitions for limitations. What is the main strategy behind limitation?

Limitations-Section has been corrected to your recommendations.

  1. It is necessary to include additional information for Figure 5.

Figure 5 has been corrected according to your recommendations and has been described in the paragraph before.

  1. What is your mean by 329-331? "The modified HI and CI are easy to obtain on cross-sectional imaging depicting cardiac structures, and can be calculated by acquiring only 2 additional parameters (thorax diameter and position of the pulmonary valve from the midline) only"

We only have to obtain 2 additional variables from cross-sectional imaging to calculate pulmonary valve postion. This has been corrected and better described.

  1. To improve your related works, 30 references are not enough and consider the following related high-quality papers in your literature review:

Thank you for your additional literature recommendations. All References hav been cited in the Introduction- and Discussion-Section next to additional references added.

The authors deeply regret, that the doctoral dissertation (Cardiac MRI in the evaluation of acute cardiac toxicity following cancer therapy, with a focus on radiation) wasn´t available so that this reference couldn´t be added. The paper now comprises 43 references.

- Cardiac MRI in the evaluation of acute cardiac toxicity following cancer therapy, with a focus on radiation (Doctoral dissertation, UNSW Sydney).

- Cardiovascular autonomic function and baroreflex sensitivity in drug-resistant temporal lobe epilepsy. Epilepsy & Behavior. 2023 Jan 1; 138:109013.

-  Heart rate and heart rate variability in patients with chronic inflammatory joint disease: the role of pain duration and the insular cortex. BMC Musculoskeletal Disorders. 2022 Dec;23(1):1-2.

-  Cardiac Phase and Flow Compensation Effects on REnal Flow and Microstructure AnisotroPy MRI in Healthy Human Kidney. Journal of Magnetic Resonance Imaging. 2022 Nov 18.

 -  . Comprehensive presurgical functional MRI language evaluation in adult patients with epilepsy. Epilepsy & Behavior. 2008 Jan 1;12(1):74-83.

-  Developing a novel integrated generalised data envelopment analysis (DEA) to evaluate hospitals providing stroke care services. Bioengineering. 2021 Dec 10;8(12):207.

- Cardiovascular computed tomography in pediatric congenital heart disease: A state of the art review. Journal of Cardiovascular Computed Tomography. 2022 May 6.

 -  A novel hybrid parametric and non-parametric optimisation model for average technical efficiency assessment in public hospitals during and post-COVID-19 pandemic. Bioengineering. 2021 Dec 27;9(1):7.

-   Neural correlates of heart-focused interoception: a functional magnetic resonance imaging meta-analysis. Philosophical Transactions of the Royal Society B: Biological Sciences. 2016 Nov 19;371(1708):20160018.

-  An integrated artificial intelligence model for efficiency assessment in pharmaceutical companies during the COVID-19 pandemic. Sustainable Operations and Computers. 2022 Jan 1; 3:156-67.

Reviewer 2 Report

Authors are proposing a modification of established MRI indexes for the assessment of pectus excavatum, relating them with the position of the pulmonary valve in cross-sectional images as a marker of cardiac lateralization. The article has proof of concept design and is based on proper methodology and appropriate statistics.

My only objection is the lack of precision in describing the steps to calculate modified Haller and Correction indices, as well as the corresponding drawing in Figure 1. The authors should make, for a broader reading public, better clarification of baseline measures needed to be taken in an improved graphical form (Figure 1), and preferably use the data from Figure 1 as an example of calculation at Figure 2. Besides, I am of the opinion that Figure 1 is not representative of pectus excavates pathology (in such a presentation Correction Index is presumably not applicable). To the careful reader, the explanation offered in the text are relatively straightforward, but a graphical representation and a working example are lacking.

Having this in mind, to my understanding (sorry if I am wrong), the statement on line 329-331 “only 2 additional parameters (thorax diameter and position of the pulmonary valve from the midline) only” should be corrected with “only 2 additional parameters (thorax diameter AT THE LEVEL OF PULMONARY VALVE, and position of the pulmonary valve from the midline) only”.

A minor typographic mistake at line 92: MRI instead of MRT.

Author Response

Dear Reviewer,

thank you very much for your highly constructive comments, we tried to consider all your remarks and found them very helpful to further enhance the quality of this paper.

Thank you very much for your thorough revision.

Please find enclosed our comments.

My only objection is the lack of precision in describing the steps to calculate modified Haller and Correction indices, as well as the corresponding drawing in Figure 1. The authors should make, for a broader reading public, better clarification of baseline measures needed to be taken in an improved graphical form (Figure 1), and preferably use the data from Figure 1 as an example of calculation at Figure 2.

We have rewritten and hopefully improved the paragraph regarding the calculation of the modified Haller and Correction-Indices. Additionally, an improved Figure was inserted with corresponding drawings. The patients Data were presented as an example of calculation.

Besides, I am of the opinion that Figure 1 is not representative of pectus excavates pathology (in such a presentation Correction Index is presumably not applicable). To the careful reader, the explanation offered in the text are relatively straightforward, but a graphical representation and a working example are lacking.

The former patient suffered from an inferior pectus excavatum. The now presented patient is hopefully more applicable.

Answer: This has been corrected by inserting a new and improved graphic and the use of variables. An additional flowchart as working example has been added.

A Flowchart regarding Data achievement has been added.

Having this in mind, to my understanding (sorry if I am wrong), the statement on line 329-331 “only 2 additional parameters (thorax diameter and position of the pulmonary valve from the midline) only” should be corrected with “only 2 additional parameters (thorax diameter AT THE LEVEL OF PULMONARY VALVE, and position of the pulmonary valve from the midline) only”.

You are totally right. This has been corrected according to your recommendation.

Round 2

Reviewer 1 Report

The authors answered all my concerns and this version is available for publication.